# A Quantitative Study on Employees’ Experiences of a Support Model for Systematic Work Environment Management in Swedish Municipalities

**DOI:** 10.3390/ijerph20054010

**Published:** 2023-02-23

**Authors:** Sofia Paulsson, Therese Hellman, Magnus Svartengren, Fredrik Molin

**Affiliations:** 1Department of Medical Sciences, Occupational and Environmental Medicine, Uppsala University, 752 37 Uppsala, Sweden; 2Uppsala University Hospital, Occupational and Environmental Medicine, 751 85 Uppsala, Sweden; 3IPF, The Institute for Organizational and Leadership Development at Uppsala University, 753 20 Uppsala, Sweden

**Keywords:** work environment, systematic work environment management, occupational health, employee participation, influence

## Abstract

Today’s working life is constantly changing, and work environmental risk factors can alter swiftly. Besides the traditional physical work environment risk factors, somewhat more abstract organizational and social work environment factors also play an ever-increasing role, both in preventing and causing work-related illness. This requires a preventive work environment management that can respond to rapid changes, and where the assessment and remedies rely more on employee participation than on predetermined threshold limits. This study aimed to investigate if the use of a support model (the Stamina model) for workplace improvements could render the same positive effects in quantitative measures that have previously been shown in qualitative studies. Employees from six municipalities used the model for 12 months. They answered a questionnaire at baseline and after six and 12 months, to detect any changes in how they characterized their current work situation and perceived their influence, productivity, short-term recovery, and organizational justice. The results showed that employees felt more influential in work situations related to communication/collaboration and roles/tasks at the follow-up compared to the baseline. These results are consistent with previous qualitative studies. We found no significant changes in the other endpoints. The results strengthen previous conclusions, namely that the Stamina model can be used as part of inclusive, modern, and systematic work environment management.

## 1. Introduction

Working life has changed dramatically over the past decades [1], affecting the panorama of work-related ill-health. While workplace accidents have decreased by 25% over the last ten years [2], mental disorders have become the main cause of sickness absence in a number of economically developed countries, including Sweden [3]. Organizational and social risk factors are increasingly common causes of mental and physical work-related illnesses [4]. Preventive measures are needed to counter-act this trend. Several reviews have shown that having influence at work is one of the most important factors in employee’s mental health [5,6]. Methods that support participation and influence at work are therefore highly interesting. This study investigated the use of such a method in six Swedish municipals. Municipalities are of particular interest, as the public sector is one of the risk sectors for work related mental illness [3].

For more than three decades, European employers have been required to conduct preventive work environment management as part of their daily operations. This means implementing systems to identify and assess risks at work, and implementing measures aimed at reducing them [7]. In Sweden, national legislation specifies that employers must integrate what is called Systematic Work Environment Management into their day-to-day activities [8]. The term “systematic” refers to this being a constantly ongoing process (Figure 1).

Traditionally, systematic work environment management has focused on identifying physical risks, making risk assessments based on established threshold limits and taking action to reduce the exposure towards a zero value. In contemporary working life, aspects such as how work is organized, employee influence, and social interaction are increasingly important for workers’ health [1,9]. This calls for new approaches regarding how to identify, assess, and minimize risks in preventive work environment management, which, in itself, promotes participation, influence, belonging, joint reflection, and communication [1]. Both participatory approaches and employee influence have been shown to be crucial ingredients for the success of workplace interventions [10,11] and to improve workers’ well-being [12]. Participatory approaches have a built-in adaptability to the context and needs of the workplace, job tasks, and workers [10], keeping the intervention constantly updated. Participatory approaches are closely connected to influence at work, which is considered one of the most important psychosocial working conditions for employees’ mental health [5,6].

Participation and influence can be linked to the control dimension in the Karasek and Theorell job-strain model. This model has its roots in the late 1970s [13,14] and plays a prominent role in occupational health research. Simplified, the model describes how job demands (e.g., workload, work pace) should be placed in relation to job control (e.g., influence on task solving) [14]. The model was later supplemented with the dimension social support [15]. If the demands exceed the perceived control, job-strain occurs, which has been linked to several adverse health effects [14]. Later studies showed that the control dimension in particular (including the concept of influence) seems linked to mental health [6,16,17]. The vast body of knowledge about control and influence at work is based on quantitative studies using one of two questionnaires: JCQ (Job Content Questionnaire [18]) and DCQ (Demand Control Questionnaire [19]). In recent years, the question has been raised if such traditional scales are obsolete or if they adequately reflect the demands in modern knowledge-intensive jobs [1]. One example is the concept of work-autonomy, which traditionally has been considered as a positive resource for well-being, but where studies in contemporary contexts have shown that open-ended and self-directed work may at the same time contribute to work-family conflicts and sleep problems [20,21]. In an attempt to describe how employees in contemporary work life experience and understand the concept of influence at work, Andersen and colleagues conducted a recent qualitative study [22]. They concluded that the interviewed employees had a multifaceted understanding of influence at work and that influence mattered to them in important ways. In the analysis, three different themes of influence were detected, divided into: (1) work tasks and performance, (2) relations and belonging, and (3) identity and becoming. The second aspect of influence, related to relations and belonging, was experienced through discussion and mutual exchange of views and knowledge in dynamic processes with co-workers and managers. These thoughts about how influence is something that is shaped and formed in relation to others in the workplace evoke concepts such as “voice behavior”, defined as employee’s sharing ideas, information, and thoughts on improvement of work tasks and the organization [23]. Another concept of “tied autonomy” was suggested by Väänänen and Toivanen (2018), to describe the fact that contemporary employees have a high level of individual freedom to make decisions and plan their work, but at the same time are dependent on co-workers, who also have a high degree of influence. Employee influence at work is, therefore, embedded in multiple social and organizational relationships. This leads back to the need for interventions that create conditions for workers’ shared voice and participation [12].

In this study, the Stamina model (further explained in the section Materials and Methods) was used to support managers to facilitate employee participation and influence on workplace improvement. The model addresses the individual employee’s experience of the work situation in an exploratory manner, without predetermined questions, and where the employees themselves decide which topics they want to cover. The emphasis of the model lies in the group’s joint reflection, choice of a common goal, and responsibility for improving the everyday work situation. The model can be described as an exploratory participatory support model, to be used as part of preventive work environment management.

The model builds upon the Model of Integrated Group Development, a model that describes how groups achieve maturity and develop skills such as dependency, counter-dependency, and trust over time [24,25]. The Stamina model includes a measure, Human Resource Index (HRI), for giving rapid and recurrent feedback to the participants and first-line managers. Previous qualitative studies on the Stamina model showed positive results regarding its implementation and use among employees and managers in Swedish municipalities [26,27,28,29].

Besides influence at work, another important aspect of the social work environment, is perceived fairness in the organization [30]. Fairness is often referred to as “organizational justice”, a concept than can be divided into four dimensions: distributive justice, procedural justice, relational justice, and informative justice [30,31,32]. In the present study, the focus was on relational justice, which emphasizes the superior’s relationship to the employees. Relational justice is associated with health in the workplace [33], and mental illness specifically [30,34].

In stress prevention, recovery is an important factor that is negatively correlated with emotional exhaustion [35], by giving greater resilience to challenges in everyday life. Quality of sleep is an important part of recovery and is connected to feeling refreshed when waking up [36]. Psychosocially favorable variables at work such as organizational justice, social support, and control have been linked to less sleep disturbances among employees [37]. On the contrary, there is a negative association between sleep quality and stress [36,38]. Impaired sleep quality is linked to adverse health outcomes, such as sick absence, diabetes, and cardiovascular disease [39].

Sickness absence is frequently used to describe reduced working ability. To capture earlier signs of reduced working capacity, e.g., due to illness or work environment-related factors that hinder the individual from performing at full potential, other aspects of production loss can be of interest. This type of production loss at work is also referred to as presenteeism [40]. Previous studies showed that the employee’s perception of their production loss can be used to screen organizations regarding production loss related to the work environment [41]. It has been suggested that this measure can function as an outcome measure for interventions aimed at improving the organizational work environment [42].

The present study aimed to examine if quantitative measures captured any perceived changes in the work environment, which had previously been described using qualitative measures among employees within Swedish municipalities, using a model that focused on employee participation and influence. Quantitative measures have the advantage of being easier to follow over time by management and employees in real-life settings.

## 2. Materials and Methods

This study was carried out as part of a larger project, focusing on the use and implementation of the Stamina model in various municipalities across Sweden, which was thoroughly described in a protocol study [43].

### 2.1. Study Design and Settings

The current study was conducted in six municipalities located in the southern and middle part of Sweden. These six municipalities were included in the study after the management had participated in a pre-program focused on establishing commitment, preparation, and planning, and then decided to use the Stamina model for a period of two years, which is considered the minimum time frame for changes to occur in this context [44]. An important perspective of the whole project was to use the model in a real-world context, mainly administrated by the municipalities themselves. This, for example, allowed the participating municipalities to choose the number of employees included in the project. Another consequence of this principle was that the questionnaires were delivered and data gathered through a web-questionnaire that was distributed via an open link. Participants were invited to take the online survey by their supervisor via e-mail. Employees were informed that participation in Stamina involved participation in a research project where all data were handled anonymously and could not be traced to specific individuals (details of the online survey can be found in Appendix A).

No individual data were gathered, only data on a group level, stating the results of a number of participants in each work group. This is in line with other employee surveys aimed at business development, due to anonymity aspects. This, however, made repeated measures on an individual level impossible. Since the permissive framework allowed the municipalities themselves to redefine how the groups were divided during the course of the study, it also became difficult to follow the results in certain work groups over time. All in all, this led to all municipalities’ data being summed into one total result.

### 2.2. Stamina Model

Stamina is an acronym for “structured and time-effective approach through methods for an inclusive and active working life” and refers to the perseverance required for continuous improvement work. The model builds upon the Model of Integrated Group Development [24], with a participatory systematic approach and has been thoroughly described elsewhere [43]. In short, the model has a specific structure that brings support and stability, but at the same time allows for a high degree of employee participation through flexibility regarding the content and actions. The starting point is the gathering of anonymous reflections about the current work situation from every employee. Respondents can enter an unlimited number of reflections, and the employee then rates each reflection as positive/negative (valence) and as influenceable or not. By using an algorithm, each employee receives a total individual score on how the work environment is graded on a scale of 0 to 100 (where 0 is as negative/not influenceable as possible and 100 is as positive/influenceable as possible). The individual score is not communicated to the employee or to the manager. The employee tags each reflection with one of nine pre-defined categories. The category design was inspired by Wheelan’s theories of group development [24]. The individual scores are summarized into the group’s score, called the Human Resource Index (HRI), also ranging from 0 to 100, and a report is generated for each group. The report serves as working material to support the work group’s reflections and discussions during a subsequent workshop, where problems and suggestions are prioritized and formalized in an action plan. There is also an integrated system for issues targeted in other parts of the organization (secondments). Figure 2 shows the workflow, where every workshop is followed by two follow-up-sessions. On each occasion, a new HRI is calculated and reported, as feedback to managers and employees. The model mimics the mandatory steps of the systematic work environment management described in Figure 1, through investigation (open web-question), risk assessment and action plan (created by the group during the workshop), and built-in sessions for follow-up. A key element is that the first-line manager oversees the process but is instructed to let the group lead the discussion, set up priorities and an action plan, and assign tasks. The implementation process has been thoroughly described in several previous publications [26,27,28,45,46].

### 2.3. Measures

#### 2.3.1. Human Resource Index

The Human Resource Index (HRI) measures employees’ perceptions of their current work situation on a scale of 0 to 100. The index is created using an algorithm where each employee receives a total individual score based on how they rate every free-text answer they use to describe their current work situation, where 0 is as negative/not influenceable as possible and 100 is as positive/influenceable as possible. The individual scores are summed into a group score, also ranging from 0 to 100. This has been thoroughly described in a previous protocol study [43]. The index has been shown to predict the risk of negative health outcomes [47].

#### 2.3.2. Perceived Productivity

The employees’ experience of their own productivity was measured using validated questions that capture the effect of health problems and work-related problems on work performance [48]. Employees were asked to answer the question: “Over the past 7 days, have you experienced any health-related problems while at work? Health problems refer to any physical or emotional problems or symptoms”. Response options were either “yes” or “no”. Health-related production loss was then measured using the question: “During the past seven days, how much did your health problems affect your performance while you were working?”. Response options ranged from 0 to 10, where 0 = “Health problems had no effect on my work” and 10 = “Health problems completely prevented me from working”. Similarly, employees were asked if they had experienced work environment-related problems (“yes” or “no”) and how these problems had affected their performance on a scale of 0 to 10, where 0 = “Work environment problems had no effect on my work” and 10 = “Work environment problems completely prevented me from working”. Work environment problems were explained as: “Work environment problems refer to any physical, psychological, or social problems that might arise in the work environment”.

#### 2.3.3. Organizational (Relational) Justice

Organizational justice targets the perceived fairness in the organization and shows links to mental illness [30,34] and other aspects of health, such as long-term inflammatory markers [49], well-being [50,51], low sickness absence, and the ability to predict future ill health [34]. The concept of organizational justice can be divided into four dimensions: distributive, procedural, relational, and informative justice [30,31,32]. In the present study, the focus was on relational justice (also referred to as interpersonal or interactional justice). This sub-domain involves the superior’s relationship with their employees; how the manager handles the employees’ personal views and rights; if the employees are treated impartially, truthfully, and with kindness. There were five statements concerning relational justice, where the participants indicated to what degree they agreed or disagreed with the statements. The scale ranged from 1 to 5, where 1 = strongly disagree and 5 = strongly agree. The five subscales were then summed into an index, which is referred to in the following as the Relational justice index.

#### 2.3.4. Short-Term Recovery during Sleep

Recovery, or rather lack of recovery, has been suggested to be a likely mediator between a stressful work situation and related health effects [35,52]. As long as you are able to recover on a regular basis, you are more likely to cope with stressful exposures and less likely to become emotionally exhausted. One important aspect in recovery is a high quality of sleep [36]. Impaired sleep quality is linked to health outcomes such as sickness absence [53], diabetes, and cardiovascular disease [39]. Organizational work conditions such as high organizational justice, social support, and control are linked to fewer sleep disturbances among employees [37]. In this study, sleep quality was assessed using a single item from the Karolinska Sleep Questionnaire [54], since “feeling refreshed when waking up” is positively associated with objective sleep quality [36]. The instructions given to the participants were that they should rate whether they have experienced the following complaint during the past three months “Do not feel refreshed when waking up”, and this was assessed on a rating scale with six response alternatives, from never to always. Hence, the question about sleep was used as a proxy for short-term recovery.

### 2.4. Data Analysis

The measurements of HRI, short time recovery, relational justice, and productivity are all variables without normal distribution, which is why descriptive statistics present the mean (M), standard deviation (SD), median (Md), and interquartile range (IQR). Change over time in output measures (T1–T3, see Table 1) was analyzed using Student’s *t*-test for HRI and Wilcoxon rank sum test for categorical variables. Changes in the proportions over time were analyzed using a two-sample Z-test for equality of proportions. The significance level was set at *p* < 0.05. Due to drop-outs, the results from the fourth time point (T4) were excluded from the statistical analysis but are shown in Table 1 for transparency. R Studio (version 27–28) was used for the descriptive and statistical analyses.

## 3. Results

### 3.1. Participants

Table 1 outlines the descriptive data regarding number of participants and number of free-text answers during the study procedure.

**Table 1 ijerph-20-04010-t001:** Timepoint for data collection (period and mean number of days since baseline), number of participants at each timepoint (summarized for all municipals), and total number of free-text answers given by the participants.

	Baseline (T1)	Time Point 2 (T2)	Time Point 3 (T3)	Time Point 4 (T4)
Data collection	Spring 2017	Autumn 2017	Spring 2018	Autumn 2018
Days after T1 (mean)	-	228 days	357 days	550 days
No. of participants (all municipals)	1191	1005	558	329
No. of free-text answers in total	5 036	3688	1999	1863

### 3.2. Work Situation Categorisation, Attitude, and Perceived Influence

Table 2 illustrates the changes in the number of reflections made by the participants and how they categorized the aspects that characterize the current work situation at baseline (T1) and at the third measurement point (T3). The categories are shown in the left column (1–10) and were pre-defined by the model. The corresponding numbers in the second column show the proportion of the total amount of reflections at baseline (T1) that contributed to each category. This sums up to 100%. The same applies to T3 in the third column. At both time points, the vast majority, about one-third of the reflections, were categorized as “work environment and health”. The second most common category was “communication and collaboration”, and in third place was “Roles and tasks”. There was a significant increase in the proportion of reflections categorized as “Implementation and follow-up” (from 5.2% to 6.6%; *p* < 0.05) and a decrease in reflections categorized as “Demands and feedback” (from 2.3% to 1.5%, *p* < 0.05).

The columns on the right in Table 2 show the proportion of reflections within each category that are considered positive and influenceable, respectively. For example, 66.3% of the reflections categorized as “results and goal fulfilment” are considered to be positive at T1 and 68.4% at T3. Among the same reflections, 59.8% are considered to be influenceable at T1 and 67.3% at T3, a non-significant increase (*p* > 0.05). Work issues categorized as “Roles and tasks” are considered significantly more positive and influenceable (increases from 57.9% to 68.7% and from 50.0% to 65.0%, respectively, *p* < 0.05). In addition, issues regarding “Communications and collaboration” are considered more influenceable (increases from 66.1% to 72.8%, *p* < 0.05).

### 3.3. HRI, Perceived Productivity, Short-Term Recovery, and Relational Justice

Table 3 summarizes the change in the rest of the outcome measures, showing that there was no significant change in the human resources index, relational justice index, short-term recovery, health-related production loss, or work environment related production loss during the study period (T1–T3).

The proportion that reported health related and work environment related production loss decreased slightly (51–46% and 48–44%, respectively), but not significantly. The proportion of employees with HRI lower or equal to 50.0 decreased slightly from 31.9% at T1 to 30.1% at T3 (*p* > 0.05).

Regarding relational justice, the index showed no significant change over time (index 24.8–25.0, *p* > 0.05). The sub-domain “supervisor’s ability to suppress personal biases”, however, showed a tendency towards a small positive change (from a mean at 3.82 (SD 1.08)–3.94 (SD 1.03), *p* < 0.05).

Table 4 shows the relationship between low HRI, relational justice index, and short-term recovery at T1. There seemed to be a connection between considering the work situation as negative and difficult to affect (low HRI), being less recovered, and not considering the organization as being fair. This connection remained at T3.

## 4. Discussion

The main findings of this study of a real-life use of an explorative participatory support model for work environment improvement are that, after using the model for one year, the employees perceived increased influence regarding aspects of “communication and collaboration” and “roles and tasks”.

This is congruent with previous qualitative results showing that working according to the model changed the dialogue and communication within the work groups. Specifically, it changed from a person-oriented, and often individually focused type of criticism, towards a task-oriented communication, focusing on the group’s common goals and tasks, and increasing the employees’ understanding of their role in relation to the whole work situation [27]. The close relationship between perceived influence and one’s perspective of one’s role at the workplace was recently discussed by Andersen et al., who stated that to contemporary employees, the meaning of influence is multifaceted, where one aspect is how it can shift the employees’ self-understanding from being a passive object to an active subject [22].

The results showed no significant change in HRI, although there was a positive change in six out of nine subcomponents regarding influence and five out of nine regarding attitude, as presented in Table 2. In other words, the change to influence or attitude had not altered sufficiently to affect the HRI significantly. In no category had the attitude or influence switched sides of the 50-marker. That is, there was no shift in valence (negative/positive) regarding attitude or influence.

Similarly, perceived productivity, short-term recovery, and organizational justice did not show any significantly changes, even if the trend showed subtle positive changes in all of those outcomes. This insignificant result differed from a previous study on the model in a hospital perioperative context [29], where the HRI improved significantly from a low level, production-loss decreased significantly, and short-term recovery improved (although it was not statistically significant). One explanation for this might be the poor starting position and other contextual factors in this previously studied organization. The current study had wider and more heterogenous settings, which were not characterized by one and the same organizational work environment. It may be that, in those groups, there was a need for a change in influence and more time to get used to the concept of group reflection before any results in HRI, productivity, recovery, or organizational justice could be detected. A previous qualitative study on the model showed that some employees/groups experienced an improvement after the first session, while others did not notice any change until the last interview, two years from the start [27]. Such differences in the implementation process may be explained by many factors [44]. “Organizational readiness for change” is suggested to be one important factor for an intervention to be effectively implemented, described as “a shared psychological state in which organizational members feel committed to implementing an organizational change and confident in their collective abilities to do so” [55]. In this study context, the dynamic relationship between relations, influence, and voice behavior, as described by Andersen et al. [22], may play a role. For practicing voice behavior, it is suggested that influence might both be a pre-condition and a result. Perhaps the perception of a certain level of influence is required before the group can reflect together at a level that leads to work environment improvements large enough to have an impact on the HRI value, productivity, recovery, and organizational justice. Previous results in the project indicated that in this particular context, giving more responsibility to the employees was a new experience for many of the managers, and thus having the mandate to identify problems and solutions was often a new experience for the employees [45]. A recent study in the Swedish construction industry, where an intervention to target stress and psychosocial working conditions was co-created with different stakeholders, showed similar results. After 24 months, the primary outcome of stress had not improved, unlike aspects of role clarity [56]. The authors suggested the time aspect as one explanation, since the main improvement in role clarity took place in the second year, and that its possible effect on stress might emerge later.

One limitation of this study is that there was a substantial drop-out. This was mainly due to a reorganization in the company that managed and distributed the questionnaires. This led to concerns and a reluctance for the included municipalities. Another limitation is the explorative design, with a focus on the process and quantitative measures, but scarce knowledge of what happens in groups not using the model and limited control over the effect of other changes in the organizations. A control group design would, therefore, be preferable in future studies on quantitative data. In addition, the summation of the results from all employees and work groups may have led to a dilution of the results, where positive results from certain work groups could not be detected in the cumulative data.

## 5. Conclusions

The results in this study showed no change in HRI, perceived productivity, short-term recovery, or relational justice, possibly partly because of a shorter follow-up period than planned. Furthermore, the lack of a control group leads to uncertainty about how other external factors affected the results. More studies are needed, preferably with a control group design, a longer follow-up period, and in other business sectors.

The results of this study, using current quantitative measures, were consistent with previous qualitative results, strengthening the finding that the use of an explorative participatory support models for work environment improvement can contribute to increased influence at work in the aspects of “communication and collaboration” and “roles and tasks”. Since influence at work is one of the most important psychosocial working conditions for employees’ mental health, such a model may have a valuable role in contemporary systematic work environment management.

## Figures and Tables

**Figure 1 ijerph-20-04010-f001:**
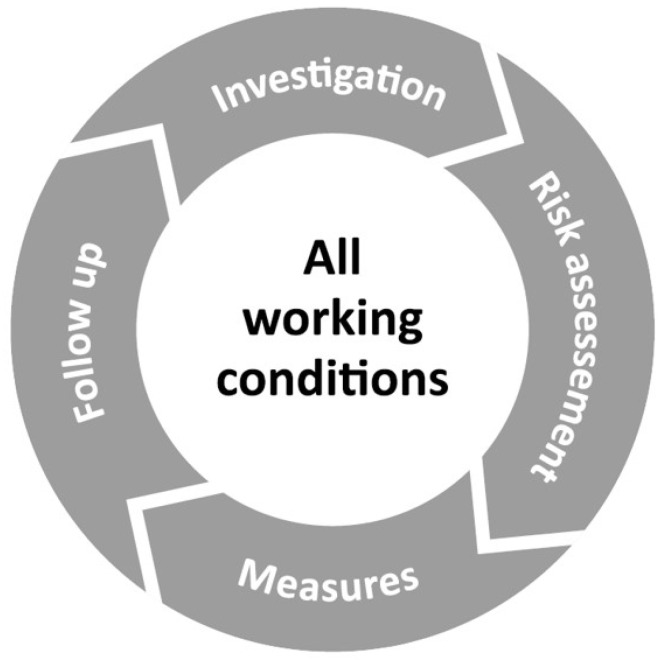
A wheel that illustrates the constantly ongoing key points in systematic work environment management (graphics from the Swedish Work Environment Authority, translated into English).

**Figure 2 ijerph-20-04010-f002:**
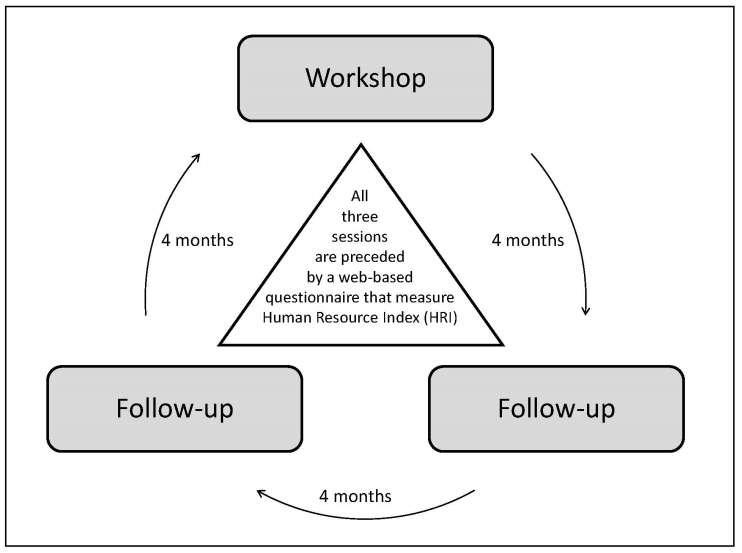
Workflow of the Stamina Model.

**Table 2 ijerph-20-04010-t002:** Respondent’s categorization and rating of free-text answers. Statistical testing was performed using a two-sample Z-test for equality of proportions (presented as *p*-value, difference).

Category	Free-Text Answers Assigned to Each Category * (%)			Positive Attitude within Each Category (%) †			Perceived Influence within Each Category (%) ‡		
	T1	T3	*p*-Value, Difference	T1	T3	*p*-Value, Difference	T1	T3	*p*-Value, Difference
(1) Results and goal fulfilment	6.4	7.6	*p* = 0.081	66.3	68.4	*p* = 0.716	59.8	67.8	*p* = 0.114
(2) External circumstances and the outside world	4.7	4.4	*p* = 0.603	41.6	39.8	*p* = 0.865	28.2	28.4	*p* = 1.00
(3) Implementation and follow-up	**5.2**	**6.6**	***p* = 0.027**	68.1	70.5	*p* = 0.712	65.0	68.9	*p* = 0.506
(4) Work environment and health	34.8	34.7	*p* = 1.00	42.7	46.5	*p* = 0.096	40.0	43.1	*p* = 0.176
(5) Roles and tasks	10.8	10.7	*p* = 0.978	**57.9**	**68.7**	***p* = 0.008**	**50.0**	**65.0**	***p* < 0.001**
(6) Skills and learning	9.0	8.2	*p* = 0.304	79.2	77.9	*p* = 0.825	73.2	72.4	*p* = 0.929
(7) Demands and feedback	**2.3**	**1.5**	***p* = 0.037**	43.0	34.5	*p* = 0.536	37.7	24.1	*p* = 0.249
(8) Time use and working methods	7.9	8.3	*p* = 0.701	45.0	38.2	*p* = 0.163	45.8	41.2	*p* = 0.371
(9) Communication and collaboration	15.8	14.7	*p* = 0.266	67.3	70.1	*p* = 0.432	**66.1**	**72.8**	***p* = 0.042**
(10) Other	3.2	3.4	*p* = 0.654	56.6	52.9	*p* = 0.717	47.2	51.5	*p* = 0.654

* Proportion of free-text answers assigned to each category. The columns sum up to 100%. † Proportion of free-text answers in each category that are considered as positive/negative (%). A proportion > 50% indicates a majority of positive attitudes towards the free-text answer. A proportion < 50.0% indicates a majority of negative attitudes. ‡ Proportion of free-text answers in each category that are perceived as influenceable (%). A proportion >50% indicates that a majority of the free-text answers are perceived as influenceable. A proportion <50.0% indicates that the free-text answer is not influenceable.

**Table 3 ijerph-20-04010-t003:** Descriptive statistics for T1 to T3 and difference between baseline and follow-up.

Variable	T1 M (SD)	T2 M (SD)	T3 M (SD)	Difference T1 and T3	*p*-Value
Human Resources Index, HRI	60.02 (20.93)	59.76 (22.70)	60.64 (23.67)	0.62	0.593 †
Relational Justice Index, RJI *	24.77 (4.42)	24.98 (4.29)	24.95 (4.40)	0.18	0.416 ‡
Short-term recovery *	3.74 (1.24)	3.81 (1.20)	3.88 (1.25)	0.14	0.253 ‡
Health-related production loss	3.12 (2.45)	3.49 (2.85)	3.30 (2.61)	0.18	0.337 ‡
Work environment related production loss	3.39 (2.43)	3.31 (2.49)	3.58 (2.54)	0.19	0.099 ‡

* Reversed items: high value indicates good relational justice and good recovery. † *t*-test ‡ Wilcoxon rank sum with continuity correction.

**Table 4 ijerph-20-04010-t004:** The relationship between HRI (individual level), relational justice, and short-term recovery at T1. n = number of employees.

	HRI <= Median n = 596 M(SD)	HRI > Median n = 595 M(SD)	Cohen’s D
Relational Justice Index	23.7 (4.78)	25.8 (3.72)	0.490
Short-term recovery	3.46 (1.23)	4.02 (1.20)	0.461

## Data Availability

The data presented in this study are available on request from the corresponding author.

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
