# Peer review of "A Quantitative Study on Employees’ Experiences of a Support Model for Systematic Work Environment Management in Swedish Municipalities"

_ijerph, 2023, doi:10.3390/ijerph20054010_

Round 1
Reviewer 1 Report
1. In line 88, "the stationary model" and "the Model of Integrated Group Development" are mentioned. Please explain these two models briefly.
2. Introduction lacks the structure settings of the paper and research contributions.
3. In line 108, six municipalities were selected for this study. Please briefly explain the reasons for choosing these in six municipalities.
4. Line 123-125,mentioned“Since the permissive framework allowed the municipalities themselves to redefine how the groups were divided during the course of the study, it also became difficult to follow the results in certain work groups over time.”
Line 146-148, mentioned“…, and a report is generated for each group. The report serves as working material to support the work group’s reflections and discussions during a following workshop, where problems and suggestions are prioritised and formalised in an action plan.”
(1) Since the team will be redefined during the research process, how can we ensure that the results of the team can continue to support the next work?
(2) (2) If it is difficult to track data, how to ensure the authenticity and validity of data? Please explain.
5. Part 2.4.1 mentions the human resource index. Please list the calculation formula of the index.
6. The conclusion is too simple. Suggest put the limitation of this study in line 333-341 into conclusion section.
Reviewer 2 Report
I reviewed this manuscript in detail, after which it was found that this manuscript should be accepted, which is suitable for this journal. Simultaneously giving the few sensible comments to authors:
1) This manuscript entirely delivered in terms of interest for readers, since the general main route of workers "participation" is limited to the consultation of workers by questionnaire, for legal purposes. This study gives another approach on workers participation, that I found very interesting.
2) The methods/approaches that have been used in this study are well described and the authors have mentioned all parameters and instruments used in detail with proper literature support.
3) Results and discussion is communicated well with appropriate justification. The main limitation of the study was identified (drop-out), as showed by Table 1 (decrease of participants at each timepoint)
4) Conclusions are supporting the research that has been carried. Maybe authors should include some perspectives of future studies on this field.
5) The manuscript has been well written.
Reviewer 3 Report
1. The link between work-autonomy ,job stress ,the potential hazards of open-ended and self-directed work are mentioned in the introduction, while preventive work environment management and the Stamina model emphasize exactly employee participation, which seems to contradict the main idea of the study. The authors are suggested to provide further explanation.
2. In the “Study design and settings” section, the study claimed “decided to use the Stamina model for a period of two years, which is considered as a minimum time frame for changes to occur in this context” (line 111-112). But the actual study data is from Spring 2017 to Spring 2018, which is only one year, and the authors are suggested to provide a reasonable explanation for this. In addition, according to Table 1 in the manuscript, the proportion of Participants who dropped out of T3→T4 was lower than that of T2→T3, and it should be reasonable to include the data of T4 in the model.
3. Other than HRI, the rationale for why the study measured Perceived productivity, Organizational (relational) justice and Short-term recovery during sleep, which are variables selected, is not well represented in the manuscript.
4. Although the study claimed that “a real-life use of an explorative participatory support model for work environment improvement are that after using the model for one year, the employees perceived increased influence regarding aspects of “communication and collaboration” and “roles and tasks.”” (line 291-294), but did the insignificance of the remaining seven pre-defined categories and HRI have a threat on the positive effect of the exploratory participatory support model? Therefore, the authors may need to give more plausible explanations for the non-significant results. (e.g. potential work stress due to open-ended and self-directed work)
5. The "preventive work environment management" and "exploratory participatory support model for work environment" mentioned in the manuscript lack sufficient theoretical explanation and logical connection.
Round 2
Reviewer 1 Report
The author has fully answered my question. The manuscript should be accepted.
Reviewer 3 Report
I recommend accepting the manuscript.